# Morphometric Characterization of the *Lidia* Cattle Breed

**DOI:** 10.3390/ani10071180

**Published:** 2020-07-13

**Authors:** Juan Manuel Lomillos, Marta E. Alonso

**Affiliations:** 1Department of Animal Production and Health, Veterinary Public Health and Food Science and Technology, Veterinary Faculty, Universidad Cardenal Herrera-CEU, CEU Universities, 46113 Valencia, Spain; 2Animal Production Department, Veterinary Faculty, University of León, Campus de Vegazana, 24071 León, Spain; marta.alonso@unileon.es

**Keywords:** *Lidia* cattle, zoometry, photogrammetry

## Abstract

**Simple Summary:**

The *Lidia* breed has great economic and social importance and is one of the most interesting breeds worldwide from a genetic point of view. The economic, social, and genetic impact has not been reflected in zootechnical studies carried out on it because the difficulty of handling and approaching of this type of animal makes measurement of live individuals almost impossible. For this reason, the first morphological characterization of *Lidia* breed was carried out using photogrammetry. This technology facilitates measurements of animals at a distance from 3-dimensional photographs. In the present work, 264 adult individuals (males and females) were studied to determine 20 morphological measurements, 5 zootechnical indexes, and individual phaneroptic information. The results show a considerable internal variability, sexual dimorphism, and the zoomometric differentiation of various genetic lines, thus improving value and complexity of this breed.

**Abstract:**

Morphometric studies in *Lidia* cattle are scarce due to the challenges of handling and approaching this breed of cattle. For this reason, the first morphological characterization of the *Lidia* breed was carried out using photogrammetry. In the present work, 264 adult individuals (184 males and 80 females), belonging to 21 different herds, were studied. A total of 20 morphological measurements and five indexes were determined in every individual. There were many positive correlations between the measures, giving the model great morphostructural harmony. Considerable internal variability of the studied parameters was observed. This breed reflected significant sexually dimorphic features and internal morphological differences between the different genetic lines of the breed. *Lidia* cattle are small and mostly have a sub-concave profile (58.4% males and 69.7% females). The male proportionality indexes and the relative depth of the thorax indicated great muscular development of the anterior third and high thoracic capacity. The phaneroptic information describes a mostly black animal with black mucous and hooves and an outstanding development of the dewlaps and the tail in the males.

## 1. Introduction

The *Lidia* cattle constitutes the most numerous autochthonous bovine breed of Spain, with 199,662 heads distributed among 917 farms, and second, in censuses, after the holstein-friesian breed [1]. The current economic and social transcendence of this livestock production is corroborated by the large number of bullfighting shows, both in our country, 19,219 shows in 2018 [2], and in the South of France, Portugal and in many regions of Latin America (Ecuador, Mexico, Colombia, Peru and Venezuela).

The origin of this breed goes back to the Middle Ages. Feudal lords raised these animals for hunting and war training purposes. The first reported evidence of artificial selection for *Lidia* bulls point to the fifteenth and sixteenth centuries. The developing farms in different areas of the Iberian Peninsula were clearly orientated to their use in taurine celebrations [3].

The modern *Lidia* breed can be considered the result of continuous selective pressure for behavioral phenotypes. Since the beginning of the 18th century, farmers used behavioral tests to choose certain ethological characteristics. Desired behaviors included prolonged durations of contest-behaviors.

These sequences of behaviors included charging against objects, people or animals, considering the morphology of the animal as secondary. Breeders deemed these patterns of behavior “*bravura*”, indicating bravery because instead of fleeing, the animal displays strength and fierceness while facing danger. Therefore, from the zootechnical point of view, it is a unique breed in the world that has a valuable morphology and genetic resource, the latter, widely studied [4,5,6,7]. However, the desired behaviors also make the modern *Lidia* animals challenging to study. Handling and restraining these cattle are very dangerous due to their surly disposition, which may have prevented the realization of traditional manual morphological studies. Nonetheless, morphological characterization is important for the conservation and biodiversity of cattle breeds [8]. By studying the phenotypes within a breed, it is possible to improve diversity and selection for adaptation to the environment and functionality [9,10].

Visual methods provide a non-invasive mechanism to evaluate cattle but are currently limited because the work requires skilled observers, is very laborious, and precision requires a high sampling rate [11,12,13,14,15,16]. Researchers previously described morphology and phaneroptics of this breed; however, few studies focused on zoometrical measures in *Lidia* cattle [17,18,19]. In addition, previous research was focused on the biometry of the horns rather than all of the features [20,21,22].

Previous zoometric-tool sets included standard measuring sticks, non-elastic measuring tape, a compass, goniometers, and calipers, which [23] required handling and restraint. In addition, handling and restraint may cause distress. Very temperamental animals may need to be immobilized using cattle crushes or containment boxes and possibly tranquilized. These procedures increase the risk of injury for both the animal and humans involved. Over the last two decades, technology advanced to improve various image-analysis techniques. One of them, known as near object photogrammetry, allows for morphological measurements on dangerous and elusive animals, such as the *Lidia* cattle, without risk to the operator or stress to the animal [24,25]. The technique was previously used to study wild animals such as elephants [26], orcas [27], and Wedded seals [28] and was validated for pigs, horses, and dairy cows [29,30,31]. The objective of this work was to adapt the photogrammetric technology to the *Lidia* breed and carry out the first zoometric and morphological characterization of the breed.

## 2. Materials and Methods 

The morphometry and phaneroptics data of 264 adult individuals (4–6 years) of *Lidia* cattle (184 males and 80 females), belonging to 21 herds, selected as representative herds in the purity of the 15 genetic lines (called “*encastes*”) that are currently preserved of the breed: *Miura, Pablo Romero, Veragua, Murube, Santa Coloma-Buendía, Santa Coloma -Graciliano, Gamero Cívico, Conde de la Corte, Atanasio-Lisardo, Domecq, Torrestrella, Núñez, Albaserrada, Vega-Villar and Navarra* [6] were enrolled. Birth records provided by the farmer were used to determine the age of the animals, which correlated with their ear tag.

All farms were located throughout the entire national territory of Spain (from Navarra in the north to Andalusia in the south), all of them in a common ecosystem called “Dehesa” or Mediterranean forest, characterized by a wooded grassland (30–100 trees per ha), mainly dedicated to extensive livestock with a Mediterranean climate with significant summer drought [1,11].

The equipment used (Figure 1) is an adaptation of the one described by Gaudioso et al. [32] that consists of a rigid and tubular structure supported in its central part by a vertical peg (Figure 1). Three cameras were attached to the structure using articulated supports. These supports allowed for adjusting the orientation of the cameras according to the size of the object to be measured and the average distance to which it is located. The lateral cameras were synchronized and remotely controlled by cable.

To minimize the risks for operator and disturbance of the animals, the photographs were taken from different angles and perspectives in the animal’s own habitat, from a distance of 10 to 15 m, without altering the normal behavior or posture. For the subsequent morphometric analysis, photographs were transformed into three dimension files using software (PhotoModeller Scanner) that allows for the possibility of performing 20 body measurements as can be seen in Figure 2, following reviewed literature standards [33,34,35].

In addition, complementary phaneroptic variables were recorded. The profile and horns conformation, layer, pigmentation of mucous membranes and hooves were measured. Also, the development of the dewlap (very developed when the dewlap reaches the carpal joint and little developed when it ends in the sternum) and tail (very developed when the tail reaches the ground, little development when it reaches the hock) were recorded. 

In addition, the following zoometric indexes have been calculated [34,36,37]:-Cephalic index = (head width/head length) × 100.-Proportionality index = (height at the withers/body length) × 100.-Relative depth of thorax index = (back-sternal diameter/height at withers) × 100.-Posterior foot index = (height at hock/height at tail) × 100.-Relative thickness of cannon bone index = (perimeter of the cannon/height at withers) × 100.-Saddling index = (height at withers + height at rump/2) − height at loins.

A descriptive statistical analysis of the variables was carried out, considering gender, for all the animals. Normality was verified using the Kolmogorov–Smirnov test. In turn, the data were subjected to a Student t-test for an independent samples test. Pearson correlation coefficients were estimated to analyze the relationships between the studied parameters. Chi-square tests (Χ^2^) were used for the phaneroptic measurements determined. These statistical analyses were done with the SPSS^®^ version 19.0 package for Windows. Finally, a principal component analysis (PCA) was performed on the morphometrical data of the males selected by the breeders as representatives of the morphology of their genetic line (*encaste*). PCA was carried out using The Unscrambler (Camo Analytics AS. Oslo, Norway) version 11.0 software under the UEX Department of Animal Production and Food Science License. This software of chemometric origin in addition to develop models validate how the model will behave in the future in the face of new cases. The following variables were used in the PCA: exterior length of the horn, head length, head width, height at withers, height at loins, height at rump, height at tail, height at shoulder, hock height, back-sternal diameter, back length, trunk length, rump length and body length. The explained variance for principal components (PC) is expressed in % and is calculated from the residual variance as:
VExp(0)=0
$$V\mathrm{Exp}(a)=\left\{\begin{array}{cc} \frac{V\mathrm{Res}\left(a-1\right)-V\mathrm{Res}\left(a\right)}{V\mathrm{Res}\left(0\right)} & \mathrm{if}\;\left(V\mathrm{Res}\left(a-1\right)-V\mathrm{Res}\left(a\right)\right)>0\\ , & a=1\ldots A\\ 0 & \mathrm{if}\;\left(V\mathrm{Res}\left(a-1\right)-V\mathrm{Res}\left(a\right)\right)\leq 0 \end{array}\right.$$
where *V*Res is any of the residual variance matrices listed in individual residual variance calculations and *V*Exp is the corresponding explained variance matrix. 

Accumulative explained variances for principal components (PC), also expressed as a percentage, are computed according to the following equation where *VExp,cum* is accumulative explained variances and *V*Res is any of the residual variance matrices listed in Individual Residual Variance Calculations:


$$VExp,cum\left(a\right)=\frac{V\mathrm{Res}\left(0\right)-V\mathrm{Res}\left(a\right)}{V\mathrm{Res}\left(0\right)}$$


## 3. Results

The twenty morphological parameters were first evaluated (Table 1), which included means, standard deviations, distributions, and Sexual dimorphism quotient SD (male/female). The exterior length of the horn was less variable in males than females (Table 1). In addition, compared to females, males had greater head width and length, horizontal and vertical diameter of the horn, back-sternal diameter, trunk, back, rump and body length, height at withers, loins, rump and height at tail. The dimorphism differences were corroborated numerically (Table 1); male body measurement/female body measurement (m/f) presented a global mean value of 1.1, indicating bulls’ morphology predominance over *Lidia* cows.

Table 2 showed the mean, standard deviation, and maximum/minimum values for the six morphometric indexes. Significant sexual dimorphism was observed in all the indexes. *Lidia* males were mesocephalic (50.6), while females were slightly dolicocephalic (44.6). The relative depth of the thorax index was significantly higher in the males as well as the relative thickness of cannon bone index. Posterior foot index and proportionality index were higher in the females. Finally, the males studied presented a marked degree of deviation of the thoracolumbar line compared with the *Lidia* cows, registering a saddling index of 4.8 cm.

Table 3 shows the linear correlations between the 20 variables analyzed in the males and females sampled. Horn morphometric variables were significant and positively correlated both in males and in females with higher values in the bulls. All the height variables were correlated, and the highest correlation value was obtained between height at rump and height at tail, 0.96 in males and 0.82 in females. Both genders presented a negative correlation between sternum height and back-sternal diameter (females −0.77 and males −0.73). Males had 59% of their parameters positively correlate, showing a harmonic morphostructural model. Females had slightly fewer harmonic features (40% parameters correlated).

Table 4 shows the percentage of different categories of morphological and phaneroptic variables studied and the results of the chi-square test depending on the gender. Studied males presented significantly less straight and more saddleback thoracolumbar lines than *Lidia* cows. Dewlap was little and developed in females and developed or very developed in males. Development of the tail was more evident in the males than in the females. No differences were observed between genders in coat and mucous membranes coloration (predominantly black), profile (mainly subconcave), and type of horns that were procerus (e.g., growth above the nape forward and curved in its medial and distal part, giving rise to crown forms and hook).

Finally, a principal component analysis has been made on 15 morphometric variables analyzed in males classified according to their genetic line. Variables that have no influence or weight were not included in the model. Males (*n* = 184) are represented spatially in Figure 3. Combining information presented in Table 5 and Figure 4, it could be observed that the PC1 (45% of explained variance, Table 6) is determined mainly by the measurements of heights in the negative quadrant and the exterior length of the horn in the positive but with less influence. Two variables clearly conditioned PC2 (10% of explained variance, Table 6)): sternum height and back-sternal diameter (Table 5 and Figure 4).

## 4. Discussion

In general, the results obtained from body dimensions place the *Lidia* breed as mid-sized compared across the bovine species [34]. For the present study, significant parameter differences (Table 1 and Table 2) corroborates the marked sexual dimorphism that exists in this breed [13]. There was clear evidence in the dimensions of the heights and body length, which indicate that the male is significantly larger than the female. In turn, significant differences among genders were observed in all indexes: cephalic, relative depth of the thorax and saddle higher in males and posterior foot and proportionality index higher in females, indicated different body proportions. The numerical estimation of dimorphism with an SD global mean value of 1.1 (Table 1) documented that the genetic selection pressure has been directed to the male morphology [38,39].

The mean height at withers observed in this study (Table 1) is less than previous means reported for *Lidia* bulls (136–143 cm) [17]. Compared with Spanish autochthonous breeds, the *Lidia* is framed in the group of mature, small-sized individuals, which includes the *cachena* (122 cm male and 117 cm female) and the *albera* (126 cm male and 121 cm female) [1,40]. Mean *Lidia* height at withers values was smaller than the rest of Spanish autochthonous breeds [1,34]. Individuals from the *Miura* line presented the highest values and the ones from *Vega-Villar* and *Navarra* lines the lowest results (Figure 3) in accordance with reviewed literature [13,41]. These results may reflect the influence of farmers´ preferences and the differences in the origin of the lines [5,6]. The sexual dimorphism of height-measurements was similar to that reported in other European cattle´s breeds, which is approximately 10 cm differences [33].

The heights of the withers, loins, rump and tail were correlated within both males and females (Table 3). This work indicates that the *Lidia* bull’s rump is slightly less than its withers (3 cm difference) giving a slightly “downhill” figure. On the contrary, female withers and rumps were level (0.8 cm difference). Males also had a marked saddling back line, registering an index of 4.8 cm. The saddling reported in the current study is similar to the previously published descriptions of this breed [13,37] and to the morphological prototype described for the rustic European bovine breeds [34]. The observed thoracolumbar line tended to be more saddled in the male (83.9%) than in the female (48.6%; Table 4).

In addition, a thorax significantly more developed was recorded in the male, endorsed in the measures of the back-sternal diameter and the length of the back, which generate significant differences between genders, (74.9 cm in the male and 64.5 cm in the female). These results state that males were selected for heavy-muscled shoulders to have a more athletic, masculine appearance than females in accordance with some authors [41]. The idea that thoracic muscle and bone development is a selected character in the male was corroborated by the index of relative thorax depth (59.5 ± 8.1 cm). All this, the results of genetic improvement, favors a great thoracic capacity that allows for greater oxygenation, leading to better performance of the bull during the taurine celebrations [5,13]. However, farmers selected males for variables of anterior third because the “*badanudo*” (developed dewlap) was a desired characteristic because it contributes to increase the visual predominance of this third. For the current study, a developed dewlap was presented in 56.2% of males, whereas in the females, 48.1% of the cases had less developed dewlaps (Table 4). 

For the present study, the body length was also shorter than that reported for the main mature Spanish autochthonous breeds [1,42] but similar to that reported for the *avileña,* also Spanish native beef cattle breed [43]. Based on the variable trunk length, the *Lidia* breed could be framed in the group of medium cattle, although presenting great variations depending on the *encastes* (results of principal component analysis are shown in Figure 3) according to literature reviewed [12,13,14,15,16]. 

On the other hand, the index of proportionality gives us a value lower than 100 for males (85.4) and females (90.6), proving that females presented a poor beef conformation improved in the bulls specially in some genetic lines such as *Pablo Romero* that has been described closer to meat production prototype breeds, with a rectangular trunk [14]. Similar values have been reported for Uruguay breeds (88.2) and *pirenaica* breed (86.3) [44,45]. Moreover, the relative depth of the thorax index provided an idea of the sarcopoietic aptitude of each breed, considering a better beef conformation for a breed when it exceeds the more from 50 points. Average values of this index in *Lidia* individuals are close to 50 (Table 2) because farmers selected muscular development in this breed focusing on the athletic appearance and performance of charging behavior during the fight [46]. These characteristics are not desired among cattle raised solely for beef production. 

The measures of height at shoulder and sternum provided information about the length of the animals’ extremities, which, in this breed, are of a medium-low size in comparison with the rest of the bovine breeds, as a selection result because breeders have been selecting bulls with less height during the last century [47,48].

The perimeter of the carpal region (31.4 cm in males) was in accordance with data published by other authors in this breed (31.2 cm) [19] but smaller than that described for other cattle breeds (33 cm for the *retinta* and 34 cm for the *berrenda*) [49]. The males’ cannon perimeter (19.7 cm), were slightly higher to values published for this breed [19,49] (18.4 cm) and considerably lower than values pointed out by other authors [17] (21–24 cm). On the other hand, this perimeter is lower than the reported for the *pallaresa* [50]; however, it is similar to that published for the *cachena* [40], both Spanish breeds. This indicates that the *Lidia* breed has a greater fineness compared to these Spanish native cattle.

The head conformation is often influenced by the origin area and environment and generally provides important characteristics of each breed [37,51]. The males in the current experiment had wider heads than the females, which generated a significantly higher cephalic index (Table 2). Head length was more variable in the males in the current study (43.3–59.2 cm in the male) than the same breed examined by Fuentes et al. [19] (44.5–52 cm) and by Barga-Besusán [17] (45–53 cm). Our results may have differed from the previous reports because we included a larger number of farms and genetic lines, which indicates that this breed may have greater morphological diversity than previously reported. Nevertheless, the means of head length obtained were lower than those described for some Spanish breeds [50,52]. The “shortened head” characteristic is often used when several authors described the *Lidia* breed [17,53]. With regard to males’ width of the head (20.3–31 cm), values presented in this study exceed the range of measurement published (21.5–28 cm) [19] and were lower than others (26–32 cm) presented in literature reviewed for this breed [17] and native Spanish breeds [50,53]. The average cephalic index in *Lidia* bulls is 50.6 and 44.6 in cows, within values considered mesocephalic and slightly dolicocephalic, respectively [12,13].

All the cattle enrolled in this study presented proceros-type horns, in the form of a hook and with different proportions and directions in their trajectory [54,55]. The results on the variety of horn pigmentation, related to the coat, with the females being significantly thin and longer (Table 1), are in accordance with the literature consulted [54,55]. The mean horizontal diameter (7.8 cm) observed was slightly larger than the vertical one (7.5 cm) describing an elliptical section of the horn as pointed out by Aparicio-Sánchez [49], which suggests a horizontal flattening of the horn. Fuentes et al. [19] reported a slightly thicker horn (8 cm) that was practically circular. The horns’ external length mean (58.5 cm) in the current project is considerably higher than that obtained by other authors thirty years ago [17], which was 46.7 cm. There was likely increased selection for bulls with larger horns because the current results are in accordance with more recent published data [21,22,56].

Considering the phaneroptic variables studied (Table 4), both genders have similar characteristics, with the black coat being predominant in the breed, not so closely followed by the brown and gray, although the female tends to a greater variability of coatings and mucous membranes and hooves’ pigmentation, as indicated by the bibliographic references consulted [13,41,54,57,58,59,60,61]. 

Generalized positive correlation between most of the parameters studied (Table 3), previously mentioned in results, supported that males of the fighting breed can be considered as representatives of a highly harmonic morphostructural model and the females of medium harmony according to consulted literature [62]. The differences between genders were probably due to increased selection pressure in the male as a result of the influence of the “*trapio*”, that could be defined as a combination of physical qualities and presence necessary for the taurine celebrations, on the males´ economic value [55,57].

Finally, the “*encastes*” *Núñez, Domecq, Gamero Cívico, Albaserrada* and *Santa Coloma* (*Graciliano* and *Buendía* lines) with a common phylogenetic provenance (called “*Vistahermosa*”) [63] were grouped in the central right part of Figure 3 in accordance with their similar morphological and genetic characteristics reported in literature reviewed [6,13] because the bulls of these *encastes* presented mean values of heights and greater exterior length of the horns.

The similarities between *Conde de la Corte* and *Atanasio-Lisardo* that come from the same livestock line, called “*Tamarón”* [63] and historically sharing their breeding area (Salamanca, Spain [52]), placed them in the left quadrants of Figure 3, spread vertically by the greater influence of the sternum height on the *Atanasio* line individuals and back-sternal diameter on the *Conde de la Corte* ones. On the other hand, differences in the morphology of animals that come from genetically isolated herds [5,6] such as the *Miura* and *Pablo Romero* lines (left above), with larger dimensions and individuals from *Vega-villar* (right below) and *Navarra* (right above) lines, could be clearly observed with a smaller size in Figure 3. The males of the “*Veragua*” line were dispersed in Figure 3 due to their lack of morphometric uniformity [13].

## 5. Conclusions

The adaptation of the photogrammetry to the *Lidia* breed allowed the first morphometric characterization obtaining a great variability in the parameters studied and values in accordance with the literature reviewed. The *Lidia* cattle breed presents smaller dimensions than other autochthonous Spanish breeds, with mostly a sub-concave mesocephalic profile. A considerable internal variability of the parameters studied has been observed, reflecting a marked sexual dimorphism and the existence of internal morphological differences between the genetic lines of the breed (“*encastes*”). A greater harmony in the male proportionality and body and muscle development compared to females could be explained by the fact that the pressure of morphological genetic selection has been exerted on the bulls due to their greater economic value.

## Figures and Tables

**Figure 1 animals-10-01180-f001:**
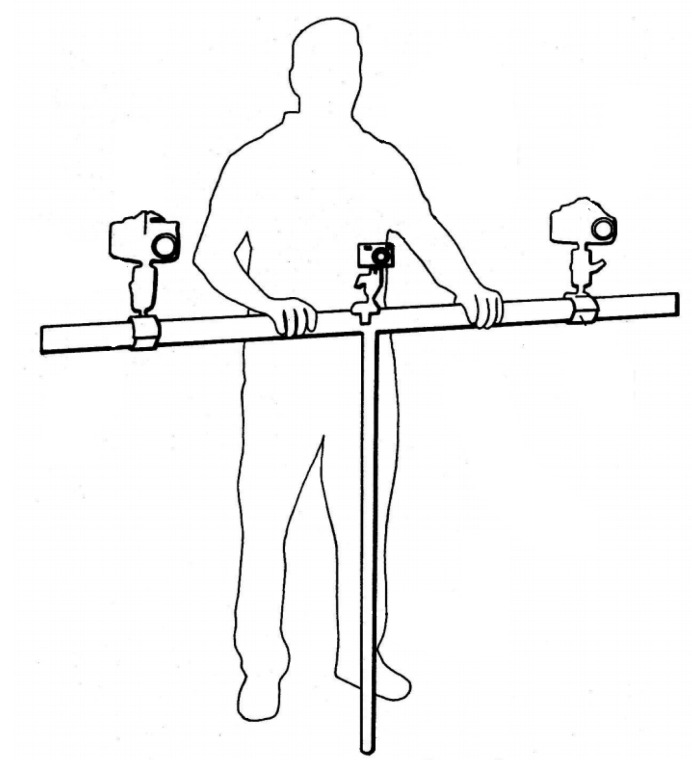
Photogrammetric equipment used in the present study.

**Figure 2 animals-10-01180-f002:**
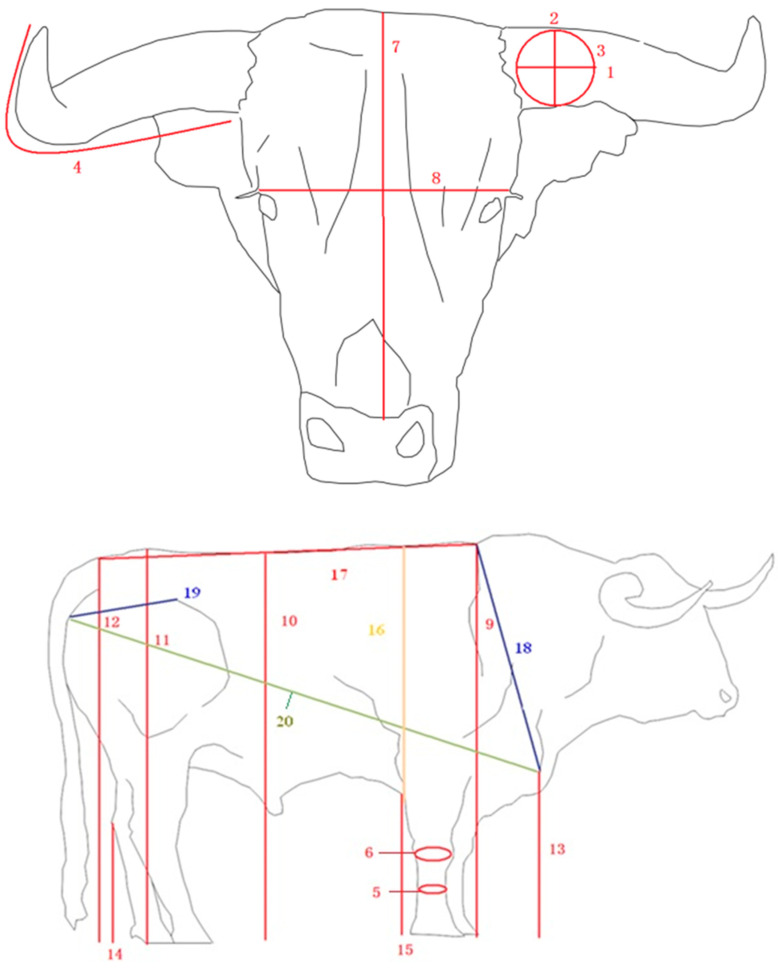
Graphical representation of the twenty body measures used in this study: horizontal diameter of the horn (**1**), horn perimeter (**2**), vertical diameter of the horn (**3**), exterior length of the horn (**4**), cannon perimeter (**5**), carpal perimeter (**6**), head length (**7**), head width (**8**), height at withers (**9**), height at loins (**10**), height at rump (**11**), height at tail (**12**), height at shoulder (**13**), hock height (**14**), sternum height (**15**), back-sternal diameter (**16**), back length (**17**), trunk length (**18**), rump length (**19**), and body length (**20**).

**Figure 3 animals-10-01180-f003:**
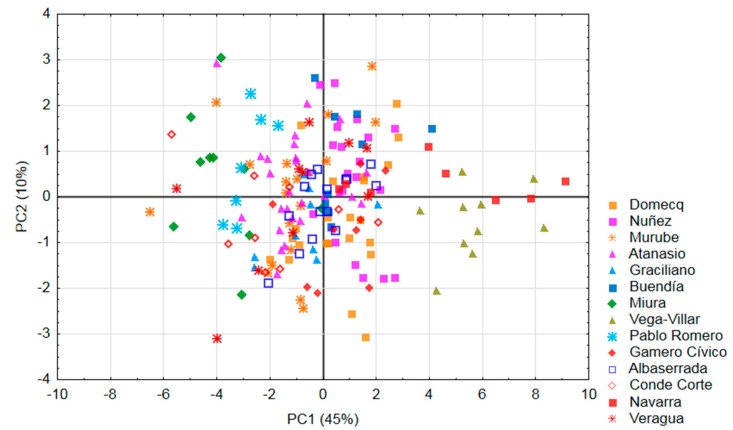
Principal component analysis on 15 morphometric variables analyzed in males (*n* = 184). Variables: Exterior length of the horn, head length, head width, height at withers, height at loins, height at rump, height at tail, height at shoulder, height at hock, sternum height, back-sternal diameter, back length, trunk length, rump length and body length.

**Figure 4 animals-10-01180-f004:**
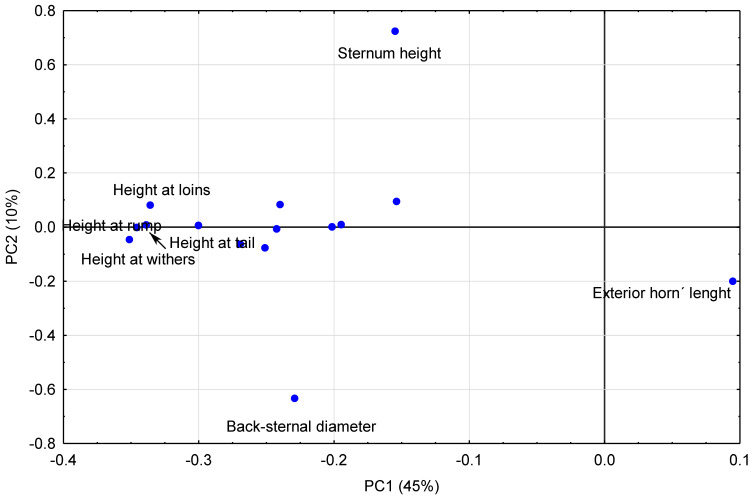
Distribution of the morphological variables for principal component 1 (PC1) and principal component 2 (PC2).

**Table 1 animals-10-01180-t001:** Average, maximum, minimum and standard deviation of studied morphological parameters and Student t-test results between genders.

Measures	Gender	Average	Maximum	Minimum	Standard Deviation	SD (m/f)
**1. Horizontal diameter of the horn**	M	7.8 ^a^	10	6	0.8	1.1
	F	6.8 ^b^	9.2	4.8	0.7	
**2. Horn perimeter**	M	25.7 ^a^	29.5	22	1.6	1.2
	F	21.6 ^b^	25.1	19.6	1.5	
**3. Vertical diameter of the horn**	M	7.5 ^a^	9.9	6.3	0.6	1.5
	F	5.1 ^b^	9	5.7	0.6	
**4. Exterior length of the horn**	M	58.5 ^a^	75	45	6.2	1
	F	57.1 ^b^	83.3	39.6	6.5	
**5. Cannon perimeter**	M	19.7 ^a^	25	16.8	1.3	1.2
	F	16.3 ^b^	21.1	14.3	1.7	
**6. Carpal perimeter**	M	31.4 ^a^	35.5	21.5	1.8	1.2
	F	26.3 ^b^	30.6	20.1	1.8	
**7. Head length**	M	49.1 ^a^	59.2	43.3	3.4	1
	F	47.1 ^b^	56.8	41.1	3.1	
**8. Head width**	M	24.8 ^a^	31	20.3	1.9	1.2
	F	21 ^b^	25.1	18.2	2.9	
**9. Height at withers**	M	127.8 ^a^	149.3	103.3	7.6	1.1
	F	116.8 ^b^	122.7	112.4	6.1	
**10. Height at loins**	M	121.4 ^a^	140.7	100.3	7.1	1.1
	F	114.4 ^b^	118.7	111.2	5.0	
**11. Height at rump**	M	124.6 ^a^	147.2	105.3	6.4	1.1
	F	117.6 ^b^	122.7	112.5	5.7	
**12. Height at tail**	M	124.3 ^a^	147.4	104.3	6.3	1.1
	F	115.9 ^b^	120.5	113.2	4.8	
**13. Height at shoulder**	M	72.7 ^a^	92	55.4	7.0	1.1
	F	68.5 ^b^	70.2	65.9	5.5	
**14. Hock height**	M	44.2	62.4	34	3.7	1
	F	43.8	44.6	41.7	3.4	
**15. Sternum height**	M	52.7	70.6	36.7	6.1	1
	F	52.3	53.4	50.7	4.4	
**16. Back-sternal diameter**	M	74.5 ^a^	97.4	58	7.6	1.2
	F	64.5 ^b^	71.4	59	6.8	
**17. Back length**	M	116.9 ^a^	144.1	99.5	7.4	1.1
	F	105.2 ^b^	111.4	101.9	8.7	
**18. Trunk length**	M	61.6 ^a^	78.1	45.1	5.6	1.2
	F	52.1 ^b^	57.4	46.8	5.6	
**19. Rump length**	M	48.8 ^a^	60.7	37.4	4.4	1.1
	F	43.1 ^b^	46.5	39	4.5	
**20. Body length**	M	150.2 ^a^	180.3	115.1	10.8	1.2
	F	128.9 ^b^	133.2	125.3	9.5	
**SD mean (m/f)**	1.1					

Different letters in the same column indicate significant differences (*p* < 0.05). Sexual dimorphism quotient SD (male/female).

**Table 2 animals-10-01180-t002:** Average, maximum, minimum and standard deviation of morphometric indexes studied and Student t-test results between genders.

Indexes	Gender	Average	Maximum	Minimum	Standard Deviation
**Cephalic index**	M	50.6 ^a^	61.6	27.3	4.3
	F	44.6 ^b^	58.1	26.8	6.2
**Relative depth of torax index**	M	59.5 ^a^	67.9	52	4.4
	F	55.2 ^b^	65.8	47	4.1
**Posterior foot index**	M	35.4 ^a^	51.8	27.3	2.6
	F	37.8 ^b^	54.7	29.8	2.9
**Proportionality index**	M	85.4 ^a^	104.1	74.7	4.6
	F	90.6 ^b^	110.3	79.8	5.7
**Relative thickness of cannon bone index**	M	15.4 ^a^	24.5	13.8	1.3
	F	13.9 ^b^	17.2	12.7	1.6
**Sadding index**	M	4.8 ^a^	7.7	2.8	3.6
	F	1.8 ^b^	2.9	0.1	2.6

Different letters in the same column indicate significant differences *p* < 0.05.

**Table 3 animals-10-01180-t003:** Pearson’s correlation coefficients matrix among the 20 variables studied in males (*n* = 184, *p* < 0.05, presented in the inferior part of the table) and in females (*n* = 80, *p* < 0.05, presented in the superior part of the table). Horizontal diameter of the horn (1), horn perimeter (2), vertical diameter of the horn (3), exterior length of the horn (4), cannon perimeter (5), carpal perimeter (6), head length (7), head width (8), height at withers (9), height at loins (10), height at rump (11), height at tail (12), height at shoulder (13), height at hock (14), sternum height (15), back-sternal diameter (16), back length (17), trunk length (18), rump length (19), and body length (20).

Variable	1	2	3	4	5	6	7	8	9	10	11	12	13	14	15	16	17	18	19	20
**1**	**1.00**	**0.39 ***	**0.32 ***	**0.28 ***	0.18	0.15	0.03	**0.28 ***	0.18	0.15	0.11	−0.09	**0.35 ***	0.14	0.04	0.01	**0.27 ***	0.11	0.17	**0.23 ***
**2**	**0.66 ***	**1.00**	**0.51 ***	**0.20 ***	−0.17	0.06	**0.31 ***	0.09	0.19	0.19	0.18	0.15	0.10	**0.21 ***	0.11	0.08	**0.29 ***	0.08	−0.12	**0.26 ***
**3**	**0.73 ***	**0.53 ***	**1.00**	0.41	−0.03	0.15	0.13	**0.25 ***	0.20	0.05	0.17	0.12	**0.35 ***	0.08	0.13	**0.26 ***	**0.20 ***	0.10	−0.18	**0.33 ***
**4**	**0.52 ***	**0.34 ***	**0.33 ***	**1.00**	0.16	−0.18	**0.21 ***	−0.04	0.03	−0.12	0.01	−0.18	**0.41 ***	−0.11	−0.04	0.12	0.18	−0.06	0.04	**0.20 ***
**5**	0.11	0.07	0.09	0.11	**1.00**	−0.13	−0.05	0,17	0.06	−0.06	0.04	0.00	0.11	0.03	0.01	−0.12	−0.03	0.13	−0.16	0.05
**6**	0.19	0.07	0.15	−0.13	**0.53 ***	**1.00**	0.12	0.15	0.18	0.17	0.11	**0.28 ***	0.13	0.15	0.13	−0.04	0.15	**0.20 ***	**0.20 ***	**0.21 ***
**7**	**0.41 ***	**0.38 ***	**0.25 ***	**0.27 ***	−0.02	0.14	**1.00**	**0.34 ***	0.12	0.18	0.08	**0.24 ***	0.09	**0.35 ***	**0.21 ***	0.18	**0.28 ***	0.11	0.13	**0.22 ***
**8**	**0.23 ***	**0.25 ***	**0.23 ***	**0.20 ***	−0.18	**0.37 ***	**0.53 ***	**1.00**	0.09	0.14	0.15	0.16	0.01	**0.39 ***	**0.27 ***	0.15	0.09	0.17	**0.32 ***	**0.07**
**9**	**0.35 ***	**0.40 ***	**0.46 ***	0.13	**0.25 ***	**0.46 ***	**0.48 ***	**0.59 ***	**1.00**	**0.58 ***	**0.40 ***	**0.57 ***	0.16	**0.38 ***	**0.20 ***	**0.39 ***	0.19	**0.21 ***	**0.40 ***	**0.33 ***
**10**	**0.25 ***	**0.29 ***	**0.28 ***	−0.02	**0.27 ***	**0.41 ***	**0.33 ***	**0.53 ***	**0.82 ***	**1.00**	**0.38 ***	**0.70 ***	0.19	**0.27 ***	**0.36 ***	**0.41 ***	**0.28 ***	**0.29 ***	**0.31 ***	**0.38 ***
**11**	**0.31 ***	**0.46 ***	**0.38 ***	0.01	0.17	**0.39 ***	**0.41 ***	**0.55 ***	**0.72 ***	**0.83 ***	**1.00**	**0.82 ***	0.12	**0.31 ***	**0.41 ***	0.28	**0.21 ***	**0.33 ***	**0.29 ***	**0.51 ***
**12**	**0.23 ***	**0.36 ***	**0.28 ***	−0.09	**0.21 ***	**0.38 ***	**0.29 ***	**0.40 ***	**0.61 ***	**0.79 ***	**0.96 ***	**1.00**	0.13	**0.42 ***	**0.40 ***	**0.30 ***	**0.28 ***	**0.34 ***	**0.37 ***	**0.28 ***
**13**	0.17	0.00	0.19	0.15	0.02	0.04	0.05	**0.29 ***	**0.35 ***	**0.32 ***	**0.23 ***	**0.31 ***	**1.00**	−0.07	0.05	0.19	−0.12	**−0.25 ***	0.18	0.15
**14**	0.11	0.12	0.18	0.09	0.12	**0.35 ***	0.12	0.09	**0.55 ***	**0.49 ***	**0.64 ***	**0.78 ***	−0.18	**1.00**	**0.52 ***	0.17	0.19	**0.22 ***	0.18	**0.26 ***
**15**	0.08	**0.29 ***	0.18	0.16	0.03	0.07	0.02	0.12	**0.59 ***	**0.64 ***	**0.52 ***	**0.68 ***	**0.38 ***	0.19	**1.00**	**−0.77 ***	0.10	0.18	0.01	**0.20 ***
**16**	0.10	0.11	0.18	0.07	−0.09	0.15	0.11	0.09	**0.60 ***	**0.48 ***	**0.37 ***	**0.53 ***	0.12	0.19	−0.73 *	**1.00**	0.12	**0.29 ***	0.18	0.18
**17**	**0.28 ***	**0.32 ***	**0.21 ***	**0.22 ***	0.15	0.09	**0.54 ***	**0.25 ***	**0.33 ***	**0.54 ***	**0.39 ***	**0.61 ***	−0.19	**0.21 ***	**0.28 ***	0.17	**1.00**	0.09	0.10	**0.33 ***
**18**	0.16	0.08	0.10	−0.09	0.16	0.12	**0.21 ***	0.10	**0.52 ***	**0.33 ***	**0.48 ***	**0.39 ***	**−0.36 ***	**0.51 ***	0.19	**0.55 ***	0.09	**1.00**	**0.58 ***	**0.30 ***
**19**	0.17	−0.14	0.11	**0.25 ***	0.07	**0.21 ***	0.03	**0.27 ***	**0.35 ***	**0.38 ***	**0.40 ***	**0.37 ***	0.11	**0.31 ***	0.19	0.13	**0.23 ***	**0.25 ***	**1.00**	**0.27 ***
**20**	**0.20 ***	0.09	**0.31 ***	**0.23 ***	**0.22 ***	**0.37 ***	**0.32 ***	0.16	**0.51 ***	**0.43 ***	**0.65 ***	**0.49 ***	0.12	**0.30 ***	**0.23 ***	0.13	**0.68 ***	**0.26 ***	0.18	**1.00**

* and bold indicates significant correlations *p* < 0.05.

**Table 4 animals-10-01180-t004:** Percentage of morphological and phaneroptic variables and chi-square test results of the animals studied depending on the gender.

Variable	Category	% Male	% Female	P
**Profile**	Straight	23.6	16.4	n.s.
	Subconcave	58.4	69.7	
	Concave	0	0	
	Subconvex	18	13.9	
	Convex	0	0	
**Horns**	Proceros	100	100	n.s.
**Thoracolumbar line**	Straight	16.1	51.4	*
	Saddleback	83.9	48.6	
**Coat**	Black	72.5	69.2	n.s.
	Red	7.7	8.4	
	Gray	9.8	7.3	
	Brown	5.5	6.6	
	Yellow	0.9	1.5	
	Mixed	3.1	5.2	
	White	0.1	0.5	
	Others	0.4	1.3	
**Mucous membranes**	Black	91.6	87.8	n.s.
	Pink	7.8	10.1	
	Mixed	0.6	2.1	
**Hooves**	Black	90.5	82.3	n.s.
	Pink	6.1	12.4	
	Mixed	3.4	5.3	
**Dewlap**	Little developed	17.8 *	48.2 *	*
	Developed	56.2 *	42.5 *	
	Very developed	26 *	9.3 *	
**Tail**	Little developed	10.1 *	26.2 *	*
	Developed	26.3 *	67.7 *	
	Very developed	63.6 *	6.1 *	

* indicates significant differences between genders (*n* = 264, *p* < 0.05).

**Table 5 animals-10-01180-t005:** Principal component (PC) eigenvalues of the 15 morphological variables analyzed in males (*n* = 184).

Morphological Variable	PC1	PC2	PC3	PC4	PC5	PC6	PC7	PC8	PC9	PC10
**Exterior length of the horn**	0.09	−0.20	0.34	−0.49	0.75	−0.08	0.09	−0.04	0.01	−0.07
**Head length**	−0.27	−0.06	0.25	0.00	−0.02	0.20	−0.24	0.67	0.12	0.28
**Head width**	−0.20	0.00	0.19	−0.45	−0.31	−0.33	−0.62	−0.09	0.16	−0.14
**Height at withers**	−0.35	−0.05	0.09	0.13	0.01	−0.11	0.13	0.06	0.21	−0.24
**Height at loins**	−0.34	0.08	0.02	−0.13	−0.05	0.04	0.19	−0.27	0.17	0.01
**Height at rump**	−0.35	0.00	−0.06	−0.17	−0.02	0.10	0.18	−0.21	0.18	0.29
**Height at tail**	−0.34	0.01	−0.10	−0.16	−0.02	0.11	0.22	−0.22	0.07	0.39
**Height at shoulder**	−0.15	0.09	0.63	0.22	−0.19	−0.25	0.35	−0.03	−0.23	−0.25
**Hock height**	−0.19	0.01	−0.38	−0.47	−0.14	−0.17	0.31	0.41	−0.49	−0.18
**Sternum height**	−0.15	0.72	0.03	0.01	0.16	−0.10	0.04	0.10	0.18	−0.12
**Back-sternal diameter**	−0.23	−0.63	0.06	0.13	−0.12	−0.03	0.10	−0.01	0.07	−0.14
**Back length**	−0.24	0.08	0.10	−0.05	0.02	0.61	−0.30	−0.29	−0.50	−0.29
**Trunk length**	−0.25	−0.08	−0.45	0.21	0.28	−0.13	−0.15	−0.03	0.23	−0.47
**Rump length**	−0.24	−0.01	−0.07	0.29	0.24	−0.52	−0.26	−0.17	−0.47	0.40
**Body length**	−0.30	0.01	0.03	0.23	0.31	0.21	−0.08	0.27	−0.05	0.03

**Table 6 animals-10-01180-t006:** Percentage of the explained and accumulated variance of the 10 principal components (PC) of the analyses.

PC	Explained Variance	Accumulated Variance
1	45.4	45.4
2	9.6	55
3	8.9	63.9
4	5.9	69.85
5	5.8	75.68
6	5.4	81.11
7	4.7	85.82
8	3.8	89.65
9	3.6	93.23
10	2.1	95.33

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
