# Peer review of "Morphometric Characterization of the Lidia Cattle Breed"

_animals, 2020, doi:10.3390/ani10071180_

Round 1

Reviewer 1 Report

11 interesting breeds worldwide from __a__ genetic point of view.
54 please include references for the fact that this is the only breed in the world which is bred for behaviour
134 A linear correlation matrix was [carried out] -> [estimated] to analyze the

Reviewer 2 Report

The work is descriptive, not research. Sex dimorphism is widely known in cattle.

Specific remarks:

Figure 2: measurements of traits 5, 6, 11, 12 and 20 are not clearly marked.

Line 134: how many groups (objects) were there in one-way analysis of variance. In my opinion, only 2 groups were compared: males and females. In this situation, ANOVA is not a valid statistical method and the F test is too weak.There are many methods for estimating correlation coefficients, what method was used here?

Line 148: "standard deviations" - this parameter is missing in tab. 1, SD in the table is described as sexual dimorphism.
Line 149-151: no feature (17) Back length.
Tab. 1. Measures - (4) Exterior length of the horn - no letters a and b in the Average column, and are incorrectly in the C.V. column
For the (19) Rump length measurement, it is unlikely that the mean M and F (48.8 and 43.1) do not differ statistically significantly.
There is no explanation for the letters a and b under Table 1.

Line 167: in table 2 there is only one statistic - Average (statistic is a function). Delete "analysis of variance" because there are only 2 groups M and F.

Table 4 lists the Chi-square test, and compares the 2-8 Category. This requires an explanation in the methodology. I think the faction test or contingency tables were used.

In lines 210-211, traits names cover the chart and legend.

There are columns PC1-PC10 on line 215 - the names of these columns are not explained.
Line 218-219 is a PC column with values 1-10 - what does this mean? The variances written there are not described in the methodology - be sure to complete them as calculated.

There are columns PC1-PC10 on line 215 - the names of these columns are not explained.
Line 218-219 is a PC column with values 1-10 - what does this mean? The variances written there are not described in the methodology - be sure to complete them as calculated.
Line 220 - explain what is designated by PC1 and PC2. What this chart shows is not described in the methodology. Can a graph on a plane be "Spatial"?
Some tables and figures are not described in the discussion, this may omit them.

Reviewer 3 Report

General comments:

The manuscript entitled “Morphometric characterization of lidia cattle breed” provides information about a significant breed from the economic and cultural point of view. Furthermore, it is well focused, with an adequate experimental design and statistic treatment.

Considering all these points, i recommend it for publication after a Minor Revision.

Specific comments:

Introduction

The introduction is well explained. The Lidia cattle plays not only a relevant role, from the economic and social point, in Spain but also in many regions of Latin America. Please consider it for this section.

I have checked the Official Regulations about Lidia breed, and I don’t understand why sometimes the first letter of the name is written in uppercase and/or lowercase. I thought that a recognized breed would be in uppercase.

Material and Methods

The experimental design has considered the different genetic lines of lidia cattle breed, and I guess the authors tried to avoid measuring animals of the same family tree.

Which was the average distance between the animal and the photogrammetric equipment? Is there an optimal distance for the measurements, which the software can correctly process it?

Results and Discussion

Figure 3 cannot be correctly observed in the PDF.

What was the average age of the animals? Because it could explain some morphological differences respect to previous studies.

Round 2

Reviewer 2 Report

In table 2, it is suggested to write 11 and 12 2 cm lower - the remaining markings are legible.

Line 218-219 is a PC column with values 1-10 - what does this mean? The variances written there are not described in the methodology - be sure to complete them as calculated.

The data of the values of the variables in each principal component (PC) were provided by the Unscrambler program as is presented directly as calculated by the program. 

In this part, the authors do not answer the question by which method the values in Table 6 were calculated, only the name of the computer program was given. This answer for mathematicians is incomplete, especially since the program given is not widely used.

I have no more comments.

Author Response

Reviewer 2

We want to thank you for your help to improve our paper, especially in the data analysis and presentation of our results.

We respond to your specific remarks below:

Figure 2: measurements of traits 5, 6, 11, 12 and 20 are not clearly marked.

We added a line to link the number of the traits and we hope that this help to mark and identify them more clearly.

Line 134: how many groups (objects) were there in one-way analysis of variance. In my opinion, only 2 groups were compared: males and females. In this situation, ANOVA is not a valid statistical method and the F test is too weak. There are many methods for estimating correlation coefficients, what method was used here?

We agree with the reviewer and following our statistical advisor (mentioned in the Acknowledge section) recommendation, we performed a t-Student test for independent samples. Now we have a lot more significant differences and we acknowledge you for giving this advice. This information is now included in Material and Methods section in Line 136.

The correlation coefficients presented in table 3 were estimated using the Pearson correlation. This information is now included in Material and Methods section in Lines 136 and 137.

Line 148: "standard deviations" - this parameter is missing in tab. 1, SD in the table is described as sexual dimorphism

Line 149-151: no feature (17) Back length.

Tab. 1. Measures - (4) Exterior length of the horn - no letters a and b in the Average column, and are incorrectly in the C.V. columna

For the (19) Rump length measurement, it is unlikely that the mean M and F (48.8 and 43.1) do not differ statistically significantly

We answer all this remarks together. We did a t Student test as previously mentioned and presented the significant results in Table 1 and Table 2. We change CV for the Standard deviation following your recommendation. We used SD (sexual dimorphism) following the recommendations that another reviewer previously did us. Lines 153 to 155 present the new significant differences in the 20 morphological parameters of Table 1and Lines 165 to 170 in the indexes.

There is no explanation for the letters a and b under Table 1.

Line 161 added * Different letters in the same column indicate significant differences p <0.05

Line 167: in table 2 there is only one statistic - Average (statistic is a function). Delete "analysis of variance" because there are only 2 groups M and F.

Table 1 and Table 2 titles changed in Lines 159-160 and 173-174.

Table 4 lists the Chi-square test, and compares the 2-8 Category. This requires an explanation in the methodology. I think the faction test or contingency tables were used.

The information of the phaneroptic variables recorded is provided in Lines 123 to 127 of Material and Methods section. Regarding the Chi-square test we followed the recommendation of our statistical advisor and information is provided in Line 139.

In lines 210-211, traits names cover the chart and legend.

This was an editing problem due to the fact that in the previous version of the principal component analysis the names of the traits were on the figure but now this information is no longer needed in the graphic representation in figure 3, because is in the figure 4 and tables 5 and 6.

There are columns PC1-PC10 on line 215 - the names of these columns are not explained.

Explanation done in Table 5 title.

Line 218-219 is a PC column with values 1-10 - what does this mean? The variances written there are not described in the methodology - be sure to complete them as calculated.

The data of the values of the variables in each principal component (PC) were provided by the Unscrambler program as is presented directly as calculated by the program. 

Line 220 - explain what is designated by PC1 and PC2. What this chart shows is not described in the methodology. Can a graph on a plane be "Spatial"?

Explanation done in Figure 4 legend in Line 229-230. “Spatial” is deleted.

Some tables and figures are not described in the discussion, this may omit them.

We are sorry to disagree with the reviewer because the figure 4 and tables 5 and 6 are necessary to better understand the results of the principal component analysis, following the indications of our statistical advisor.

This manuscript is a resubmission of an earlier submission. The following is a list of the peer review reports and author responses from that submission.

Round 1

Reviewer 1 Report

General

It is recommended to write the document in third person.

Abstract
Relevant results are missing.

Materials and methods

Indicate the place (autonomous community, province, etc.) where the study was carried out, including climate characteristics and geographic location.

In introduction it is indicated that this is the first study to be carried out in order to characterize the lidia cattle breed morphology. However, it is already well known that this breed does not constitute a homogeneous population and that there are various castes, “encastes” and lines with well-defined phenotypic characteristics. As this is the first study of its kind, there is a risk of erroneously generalizing the lidia breed morphology with the data obtained here, so it is necessary to include the characteristics (breed, line) of the 21 herds studied. If the herds belong to different genetic groups, this will later help the authors explain the variability found in some body measurements. If they belong to a single genetic group, then this should be specified in the title and throughout the document. In fact the authors justify this in line 151 of the manuscript.

How was the age of the animals determined? Indicate if the breeder's records were used

Figure 1. In this figure the measures trunk length (17) and back length (18) are inverted, they must be trunk length (18), back length (17).

Line 88: It is mentioned that tail insertion was measured as a phenotypic variable, however in Table 3 and in the discussion this variable is referred to in a context in which it seemed that what it evaluated was its length. If tail length was measured, authors should specify how a tail was visually determined to be short, medium, or long. The same suggestion applies in the case of the dewlap.

In various parts of the document (for example on lines 92 and 105) the back-sternal diameter measurement appears, which does not appear in the list of measurements taken, nor in Figure 2, Table 1 and Table 2. In fact, in Figure 2 measurement 16 which corresponds to trunk perimeter, it is indicated in a linear way and not with an ellipse as in the case of the perimeters of the forelimb. It is effectively the thoracic perimeter, or it is back-sternal diameter. Also, back-sternal diameter (line 92) or sternal back diameter (line 105) is used interchangeably. Review, correct and standardize.

Line 92: It is more appropriate to use the term Relative depth of thorax index.

Line 94: It is more appropriate to use the term Relative thickness of cannon bone index.

Was the normality of the data analyzed? Mention which test was used and show the statistic. If the above was not found and the distribution of the data is not normal, the Anova and Pearson's correlation should not be performed without the prior transformation of data. The other option is to use non-parametric statistics (Spearman correlation for example).

Results

Table 1. Very extensive, separate the zoometric indices in another table.

The description of the results tables is very poor, highlight the main findings. In fact, the text written on lines 113 and 114 of the discussion section " Black coloration of coat and mucous membranes, mainly subconcave profile, and a great development of horns, proceros type, in both males and females…….." is how Table 3 and the rest of the tables should be described.

Table 2. If 264 animals (184 males and 80 females) were studied, why does this table only show data from 186 animals? If these correlations mix female and male body measurements as it appears to be, since there are no 186 animals of the same sex, the analysis is incorrect. Data from males and females should be analyzed separately, especially considering the findings of this study that indicates the clear presence of sexual dimorphism.

Table 2. In this table the statistically significant correlation coefficients are shown in red; it is more advisable to use the * sign after the value of the coefficient to indicate statistical significance. One of the objectives that is sought in a fighting bull is its “trapío”, a characteristic that combines the physical qualities and the presence necessary for the fight. One of these physical qualities is the harmony of its morphological structure, which is obtained with the amount of significant positive correlations between its body measurements. A highly harmonic model is one in which the number of significant and positive correlations exceeding a total of 50 %, a medium harmonic model when they were closer to 50 %, and a low harmonic model when only 25 % of the measurements were significant and positively correlated. These data already have them in Table 2, why is this not estimated?

Table 3. In the profile variable, the sub-convex category is repeated, and the sub-concave category is missing. Also, for the tail variable, change the big category to long.

Table 3. Was analysis of variance carried out on the percentages obtained in the categories of the tail and dewlap variables? Explain how it was done or if it is an error, remove the superscripts from the table.

Since there is no translation for the term "procero", it is recommended to place an explanatory note the first time it appears in the manuscript.

Discussion

The discussion is very extensive and in the context of the study it does not seem very useful to compare the values of each of the body measurements obtained here against those of other cattle breeds. It is evident that there will be notable differences between the morphology of a breed that evolved and was selected for a very specific purpose such as fighting and that of the breeds that were selected to produce meat or milk. The discussion should focus on explaining the morphological variation considering the different castes or genetic lines of the herds studied, the zootechnical management implemented by each producer, the consanguinity of the herds and the selection characteristics that each breeder has sought to implement in his lidia bull (a great development of the posterior third, a great thoracic capacity that allows greater oxygenation and strong limbs leading to better performance of the bull during the “lidia”.

In the case of zootechnical indices, here it would be worth comparing them against those of a couple of meat or dairy breeds, since they provide a more complete idea of what the breed is like (cephalic, posterior foot,  and saddling) and its zootechnical purpose (proportionality, thorax depth relative and relative cannon thickness). In the case of these last indices, explain what they indicate including a reference value , for example: a value lower than 100 in the proportionality index indicates a high meat aptitude  because it shows a predominance of body length over height at withers and indicates that the shape of the body tended to be a rectangle, characteristic of meat breeds.

Lines 118-122. The authors refer to sexual dimorphism based on the statistical differences between some of the body measurements. However sexual dimorphism can be easily and numerically estimated as male body measurement/female body measurement (m/f), and the global mean was calculated as the average of all the values obtained. Each quotient greater than 1 indicates superiority of the male and each quotient less than 1 indicates superiority of the female. In some species or breeds, these values indicate on which body measurements the pressure of genetic selection has been directed. The authors could include a column at the end of Table 1 to show this information with the heading: *SD (m/f).       *Sexual dimorphism (male/female).

Conclusions

Write in an orderly way, grouping in concise sentences the main findings in the animal's phenotype, in the variability including the main bodily measures that cause it, in the correlations and of course the method used to carry out the study.

References

Reference [53] does not appear in the text.

The writing of the references does not comply with the norms of the journal.

The reference: Darmanin, N.E., Camacho, E., Molina, A., Degado, J.V., Fresno, M., 1992. Descripción 337 morfológica y zootécnica de la vaca Palmera como clave para su preservación. Arch. Zootec. 338 44: 353-360, is incorrect, the year and volume number have no correspondence. Check the journal.

Author Response

Reviewer 1:

Thank you for your thorough review and with your advice we improve the draf and corrected our errors.

We have marked in the text all the changes made underlined in blue -------------

We respond to your comments below:

General

It is recommended to write the document in third person.

We have changed all verb tenses to third person

Abstract

Relevant results are missing.

We have incorporated relevant results

Materials and methods

Indicate the place (autonomous community, province, etc.) where the study was carried out, including climate characteristics and geographic location.

The study has been carried out with the entire national territory of Spain, from Navarra in the north of Spain to Andalusia in the south, therefore, it includes a great diversity of climates. We have included in the text the description of the ecosystem where the lidia cattle is raised: the “dehesa”, which is common for all farms. In Lines 90 to 93 is presented in this way:

All farms were located throughout the entire national territory of Spain (from Navarra in the north to Andalusia in the south), all of them in a common ecosystem called “Dehesa” or mediterranean forest, characterized by a wooded grassland (30-100 trees per ha), mainly dedicated to extensive livestock with a mediterranean climate with significant summer drought [1,11].

In introduction it is indicated that this is the first study to be carried out in order to characterize the lidia cattle breed morphology. However, it is already well known that this breed does not constitute a homogeneous population and that there are various castes, “encastes” and lines with well-defined phenotypic characteristics. As this is the first study of its kind, there is a risk of erroneously generalizing the lidia breed morphology with the data obtained here, so it is necessary to include the characteristics (breed, line) of the 21 herds studied. If the herds belong to different genetic groups, this will later help the authors explain the variability found in some body measurements. If they belong to a single genetic group, then this should be specified in the title and throughout the document. In fact the authors justify this in line 151 of the manuscript.

We have entered in the text in Lines 84 to 87 :

belonging to 21 herds, selected as representatives herds in purity of the 15 genetic lines (called “encastes”) that are currently preserved of the breed: Miura, Pablo Romero, Veragua, Murube, Santa Coloma-Buendía, Santa Coloma -Graciliano, Gamero Cívico, Conde de la Corte, Atanasio-Lisardo, Domecq, Torrestrella, Núñez, Albaserrada, Vega-Villar and Navarra [6] were enrolled.

The rest of the genetic lines described in some bibliographic reference for the breed are in extinction, without a sufficient number of adult animals, or are not currently kept in purity,  have been mixed with other lines. For this reason we have only sampled the “encastes” mentioned as representatives of the breed at present.

How was the age of the animals determined? Indicate if the breeder's records were used

In order to identify each animal, the ear tag number and the number marked on fire on the animal were taken into account. In the text in Lines 88 and 89:

Birth records provided by the farmer were used to determine the age of the animals, which correlated with their ear tag.

Figure 1. In this figure the measures trunk length (17) and back length (18) are inverted, they must be trunk length (18), back length (17).

We think that it is Figure 2 (figure 1 is the photogrammetric equipment). We have change it and verified that it is correct in the others tables.

Line 88: It is mentioned that tail insertion was measured as a phenotypic variable, however in Table 3 and in the discussion this variable is referred to in a context in which it seemed that what it evaluated was its length. If tail length was measured, authors should specify how a tail was visually determined to be short, medium, or long. The same suggestion applies in the case of the dewlap.

For the tail, it was measured the “height at tail – measure nº 12” (tabla 1) and valued the “tail development” (tabla 3).  In the case of the dewlap we have only valued its development (tabla 3). We have clarified in the text (material and methods) how the development of both regions is evalued in Lines 123 to 125:

(very developed when the dewlap reaches the carpal joint and little developed when it ends in the sternum) and tail (very developed when the tail reaches the ground, little development when it reaches the hock)

In various parts of the document (for example on lines 92 and 105) the back-sternal diameter measurement appears, which does not appear in the list of measurements taken, nor in Figure 2, Table 1 and Table 2. In fact, in Figure 2 measurement 16 which corresponds to trunk perimeter, it is indicated in a linear way and not with an ellipse as in the case of the perimeters of the forelimb. It is effectively the thoracic perimeter, or it is back-sternal diameter. Also, back-sternal diameter (line 92) or sternal back diameter (line 105) is used interchangeably. Review, correct and standardize.

We have detected our multiple errors and we have corrected and standardized the term of the linear measurement carried out in our case: the “back-sternal diameter” as in Line 145 or 239.

Line 92: It is more appropriate to use the term Relative depth of thorax index.

We have replaced the term as in Line 129 or Line 259.

Line 94: It is more appropriate to use the term Relative thickness of cannon bone index.

We have replaced the term as in Line 131 or Table 2 Line 168 to 169.

Was the normality of the data analyzed? Mention which test was used and show the statistic. If the above was not found and the distribution of the data is not normal, the Anova and Pearson's correlation should not be performed without the prior transformation of data. The other option is to use non-parametric statistics (Spearman correlation for example).

We checked normality using Kolmogorov-Smirnow test. We are sorry not addres your recomendation to show statistic in the text because would made the paper excesively long. We carried it out in the same way as we to in Escalera et al. 2013. Influence of intense exercise on acid-base, blood gas and elctrolyte status in bulls. REsearch in Veterinary Science, 95: 623-628. In text Line 134 is mentioned:

Normality was verified using Kolmogorov-Smirnov test

Results

Table 1. Very extensive, separate the zoometric indices in another table.

We have separated in table 2 in Line 168:

Table 2: Main statistics of the registered indexes and analysis of variance between genders.

The description of the results tables is very poor, highlight the main findings. In fact, the text written on lines 113 and 114 of the discussion section " Black coloration of coat and mucous membranes, mainly subconcave profile, and a great development of horns, proceros type, in both males and females…….." is how Table 3 and the rest of the tables should be described.

We have work in the Results section following your advice. New text is added marked in the manuscript as in Lines 147 to 149:

The dimorphism differences were corroborated numerically (Table 1); male body measurement/female body measurement (m/f) presented a global mean value of 1.1 indicating bulls´ morphology predominance over lidia cows.

And many others.

Table 2. If 264 animals (184 males and 80 females) were studied, why does this table only show data from 186 animals? If these correlations mix female and male body measurements as it appears to be, since there are no 186 animals of the same sex, the analysis is incorrect. Data from males and females should be analyzed separately, especially considering the findings of this study that indicates the clear presence of sexual dimorphism.

There is an error in the number of animals. Initially, two different correlation tables were made by gender, the results were very similar. As the article was too long, we join the results into the same correlation table for the 20 parameters, males and females.

Following your indications, we rescued the two correlation tables for the 184 males and the 80 females: Lines 176 to 181:

Table 3: Pearson's correlation coefficients matrix among the variables studied in males (n=184, p < 0.05). Horizontal diameter of the horn (1), horn perimeter (2), vertical diameter of the horn (3), exterior length of the horn (4), cannon perimeter (5), carpal perimeter (6), head length (7), head width (8), height at withers (9), height at loins (10), height at rump (11), height at tail (12), height at shoulder (13), height at hock (14), sternum height (15), back-sternal diameter (16), back length (17), trunk length (18), length of rump (19), body length (20).

Lines 185 to 190

Table 4: Pearson's correlation coefficients matrix among the variables studied in females (n=80, p < 0.05). Horizontal diameter of the horn (1), horn perimeter (2), vertical diameter of the horn (3), exterior length of the horn (4), cannon perimeter (5), carpal perimeter (6), head length (7), head width (8), height at withers (9), height at loins (10), height at rump (11), height at tail (12), height at shoulder (13), height at hock (14), sternum height (15), back-sternal diameter (16), back length (17), trunk length (18), length of rump (19), body length (20).

Table 2. In this table the statistically significant correlation coefficients are shown in red; it is more advisable to use the * sign after the value of the coefficient to indicate statistical significance. One of the objectives that is sought in a fighting bull is its “trapío”, a characteristic that combines the physical qualities and the presence necessary for the fight. One of these physical qualities is the harmony of its morphological structure, which is obtained with the amount of significant positive correlations between its body measurements. A highly harmonic model is one in which the number of significant and positive correlations exceeding a total of 50 %, a medium harmonic model when they were closer to 50 %, and a low harmonic model when only 25 % of the measurements were significant and positively correlated. These data already have them in Table 2, why is this not estimated?

We have changed the way of marking significant correlations using the * sign after the value.

We wan to thank the reviewer for the comment and we have also added the following in Lines 171 to 174.

Several morphometric variables were positively correlated (p<0,05). Males had 60.5% (Table 3) of their parameters positively correlate, showing a harmonic morphostructural model. Females had slightly fewer harmonic features (41% parameters correlated; Table 4)

Table 3. In the profile variable, the sub-convex category is repeated, and the sub-concave category is missing. Also, for the tail variable, change the big category to long.

We have fixed the profile classification error.

For the tail, at the request of another reviewer we have considered three categories: “little developed, developed, very developed.”

Table 3. Was analysis of variance carried out on the percentages obtained in the categories of the tail and dewlap variables? Explain how it was done or if it is an error, remove the superscripts from the table.

Animals were classified in each of the exposed categories, in this case the development of tail and dewlap. With the percentages by sex a contingency table was made and finally, in order to know if the distribution in males and females showed differences, the chi-square test was performed (p<0,05). We have added all this in material and methods in Lines 194 and 198.

Since there is no translation for the term "procero", it is recommended to place an explanatory note the first time it appears in the manuscript.

Explanatory note in Lines 196 and 197.

“proceros type (growth above the nape forward and curved in its medial and distal part, giving rise to crown forms and hook)”

Discussion

The discussion is very extensive and in the context of the study it does not seem very useful to compare the values of each of the body measurements obtained here against those of other cattle breeds. It is evident that there will be notable differences between the morphology of a breed that evolved and was selected for a very specific purpose such as fighting and that of the breeds that were selected to produce meat or milk. The discussion should focus on explaining the morphological variation considering the different castes or genetic lines of the herds studied, the zootechnical management implemented by each producer, the consanguinity of the herds and the selection characteristics that each breeder has sought to implement in his lidia bull (a great development of the posterior third, a great thoracic capacity that allows greater oxygenation and strong limbs leading to better performance of the bull during the “lidia”.

We have summarized the size of the discusión, focusing on lidia breed. The problem is that there are few published studies and little morphometric information.

To enlarge the information on the different genetic lines within the lidia breed, we have carried out a principal component analysis with the morphometric data of the males, selected by the breeders as examples of the morphology of each “encaste” in the herd. We have only performed with the males because we understand that they are the individuals that are selected for their morphology and the females for their behavior, because they dont go to the shows. Furthermore, the existing bibliography on this topic has focused on males and that allow us to compare our results.

In the case of zootechnical indexes, here it would be worth comparing them against those of a couple of ances dairy breeds, since they provide a more complete idea of what the breed is like (cephalic, posterior foot,  and saddling) and its zootechnical purpose (proportionality, thorax anc relative and relative cannon thickness). In the case of these last ances, explain what they indicate including a reference value , for example: a value lower an 100 in the proportionality index indicates a high meat aptitude  because it shows a predominance of body length over height at withers and indicates that the shape of the body tended to be a rectangle, characteristic of meat breeds.

We have defined the indexes and compared with the information that we have from other bovine breeds.

Lines 118-122. The authors refer to sexual dimorphism based on the statistical differences between some of the body measurements. However sexual dimorphism can be easily and numerically estimated as male body measurement/female body measurement (m/f), and the global mean was calculated as the average of all the values obtained. Each quotient greater than 1 indicates superiority of the male and each quotient less than 1 indicates superiority of the female. In some species or breeds, these values indicate on which body measurements the pressure of genetic selection has been directed. The authors could include a column at the end of Table 1 to show this information with the heading: *SD (m/f).       *Sexual dimorphism (male/female).

We acknowledge this point that seems very interesting to us, and we added a column to Table 1 following yours instructions and also in the text the references to it as in Lines 147 to 149

The dimorphism differences were corroborated numerically (Table 1); male body measurement/female body measurement (m/f) presented a global mean value of 1.1 indicating bulls´ morphology predominance over lidia cows.

And Lines 217 to 218  

The numerical estimation of dimorphism with a SD global mean value of 1.1 (Table 1) documented that the genetic selection pressure has been directed to the male morphology [38,39].

Conclusions

Write in an orderly way, grouping in concise sentences the main findings in the animal's phenotype, in the variability including the main bodily measures that cause it, in the correlations and of course the method used to carry out the study.

we have tried to rewrite the conclusions taking into account your suggestions

References

Reference [53] does not appear in the text.

Fixed the error

The writing of the references does not comply with the norms of the journal.

We have corrected all references to the journal format.

The reference: Darmanin, N.E., Camacho, E., Molina, A., Degado, J.V., Fresno, M., 1992. Descripción morfológica y zootécnica de la vaca Palmera como clave para su preservación. Arch. Zootec. 338 44: 353-360, is incorrect, the year and volume number have no correspondence. Check the journal.

We have replaced the reference with a more current one:

Salako, A. E. 2006. Application of Morphological Indices in the Assessment of Type and Function in Sheep. International Journal of Morphology 24(1). DOI: 10.4067/S0717-95022006000100003

Reviewer 2 Report

many instances of "animals" are better served with "individuals"

43 clearly orientated to their use for **bullfighting **purpouses [3].
44 Therefore, **these days breed can be considered the result of the continuous selection work carried out
44 Therefore, these **days breed can be considered
-> days'
48 Therefore, from the zootechnical point of view, it is a unique **animal in the world that has an
-> breed
49 important and varied genetic resource,
-> not the right term for genetic variability, and enfasis should be put on the fact that no selection for morphology was ever performed on this breed and any changes are due to correlated responses. Given that it has been selected for the same set of traits for centuries now, it seem apropiate that the authors include the list at this point.
52 morphological studies due to the danger of harm for the people and **for the own animal -> the animal itself
55 By studying the phenotype of the individuals of a breed we can evaluate its diversity [9]
-> this is a rather archaic approach to genetic diversity.
59 made from a **distant visual point of view [11-16].
->
62 So far, the zohometric evaluation of the different bovine breeds has been done manually, using
63 different instruments such as standard or “Aparicio” stick, non-elastic measuring tape, compass,
64 goniometers or calibers [23],
-> inappropriate or non-existant terms: zohometric, manually is not the right word to describe such methods, Aparicio sticks (centainly not familiar to most researchers), calibers (probably the wrong term, try calipers).
64 and a **certain inaccuracy
-> certain is not the right word in this context.
71 **a first zoometric and morphological characterization of the breed.
-> the
74 80 females **(4-6 years),
-> unnecessary repetition hinders clarity
74 belonging to 21 herds, selected according to criteria of genetic purity [6].
-> selection requires further explanation: does it refer to herds or to individuals? the "criteria of genetic purity" is not solved with a reference. This point is absolutely critical as the declared objective of the paper is the description of the breed, and it might be done on a selected sample!
83 For the subsequent morphometric analysis photographs were **three-dimensionally converted
-> inappropriate or non-existant term
86 At the same time we have recorded
-> do they mean that these traits were recorded simultaneously?
88 In addition, the following **zoomometric indices
-> inappropriate or non-existant term
103 presents the most variability is the length of the horn, especially in the female (C.V of 18.94%).
109 Finally, Table 3 shows the percentage of males and females that present the categories of
-> are these significant differences? (this conclusion requires a two sample test's p-value for CV differences)
110 morphological and phaneroptic variables studied.
-> unclear
112 In general we can consider the lidia breed as animals of mesolorian format within the bovine **specie.
-> This paragraph is very badly writen, plus I don't think this is the intended meaning for the word "specie". Same for "dewlap", as its hipothesized relation with the dorsal-lumbar line requires further explanation or references.
The quality of the text degrades at this point, making it difficult to read.
131 exceeded the bulls in these study.
-> grammar
132 female of Argentinean Creole cattle,
-> unjustified reference to a different breed. The context of breeds to compare belongs to the introduction.
134 The dorsal-lumbar line is observed tending more to saddling in the male
-> grammar
149 In the case of the length of the head (43.44 - 59.17 cm in the male),
-> is this the absolute range, or is it an inter-quantil? It must be stated here. An absolute range may be affected by outliers and should be avoided. The number of decimal digits is probably excesive for the precision of the set up.
152 The average values obtained are lower than those described for the retinta breed [38], for the canarian
153 breed [39-40], for the avileña, limousin, saler and charolais breeds [41] and for the pallaresa breed
-> The authors don't provide a reason for comparing with these breeds.
-> Breeds in mixed cases, articles in mixed languages
187 brevile
-> unclear
204 The anterior part predominates in the lidia breed [33], but there are genetic lines (encastes) that
The issue with "encastes" is critical and must be dealed with much earlier in this manuscript, and with greater detail.
210 which as adults will outperform fighting animals.
-> unclea215
215 idea of the sarcopoietic aptitude of each breed, considering the meatiness of a breed the more it
-> grammar
222 borlon
-> language?
229 breeds, as a result of the selection because the breeders have been selecting over the years bulls "short-
legsl"
-> unclear
238 for the bovine cattle of Uruguay
-> language
265 indexs
-> unclear

Author Response

Thank you for your thorough review and with your advice we improve the draf and corrected our errors.

We have marked in the text all the changes made underlined in yellow: -------------

We respond to your comments below:

many instances of "animals" are better served with "individuals"

We have revised all the text and replaced 8 “animals” words for “individuals”

43 clearly orientated to their use for **bullfighting **purpouses [3].

We changed to taurine celebrations in Line 48.

44 Therefore, **these days breed can be considered the result of the continuous selection work carried out

44 Therefore, these **days breed can be considered days'

After the English revision of our Dr. Lindsey Hulbert (KSU) the text in Lines 49 and 50 is:

The modern lidia breed can be considered the result of continuous selective pressure for behavioral phenotypes.

48 Therefore, from the zootechnical point of view, it is a unique **animal in the world that has an -> breed

As in the previous sentence, after the English revision, the text is now in Lines 56 and 57:

Therefore, from the zootechnical point of view, it is a unique breed in the world that has a valuable morphology and genetic resource, the latter, widely studied [4-7].

49 important and varied genetic resource,  not the right term for genetic variability, and enfasis should be put on the fact that no selection for morphology was ever performed on this breed and any changes are due to correlated responses. Given that it has been selected for the same set of traits for centuries now, it seem apropiate that the authors include the list at this point.

Following your recomendation and after the English revision, the text is now in Lines 51 and 56:

Since the beginning of the 18th century, farmers used behavioral tests to choose certain ethological characteristics. Desired behaviors included prolonged durations of contest-behaviors.

These sequences of behaviors included charging against objects, people or animals, considering the morphology of the animal as secondary. Breeders deemed these patterns of behavior “bravura,” indicating bravery because instead of fleeing, the animal displays strength and fierceness while facing danger.

52 morphological studies due to the danger of harm for the people and **for the own animal -> the animal itself

After the English revision, the text is now in Lines 72 and 73

These procedures increase risk of injury for both the animal and humans involved.

55 By studying the phenotype of the individuals of a breed we can evaluate its diversity [9]

-> this is a rather archaic approach to genetic diversity.

After the English revision, the text is now in Lines 61 and 63

]. By studying the phenotypes within a breed, it is possible to improve diversity and selection for adaptation to the environment and functionality [9-10].

59 made from a **distant visual point of view [11-16].

After the English revision, the text is now in Lines 64 to 67

Visual methods provide a non-invasive mechanism to evaluate cattle, but currently are limited because the work requires skilled observers, is very laborious, and precision requires a high sampling rate. [11-16]. Researchers previously described morphology and phaneroptics of this breed, however few studies focused on zoometrical measures in lidia cattle [17-19].

62 So far, the zohometric evaluation of the different bovine breeds has been done manually, using

63 different instruments such as standard or “Aparicio” stick, non-elastic measuring tape, compass,

64 goniometers or calibers [23],

-> inappropriate or non-existant terms: zohometric, manually is not the right word to describe such methods, Aparicio sticks (centainly not familiar to most researchers), calibers (probably the wrong term, try calipers).

Changed in Lines 69 and 70 to:

Previous zoometric-tool sets included standard measuring sticks, non-elastic measuring tape, compass, goniometers, and calipers which [23] required handling and restraint

64 and a **certain inaccuracy

-> certain is not the right word in this context.

This is changed and not used in the new tect Lines 64 and 67 because was re-frased in a more positive way.

71 **a first zoometric and morphological characterization of the breed.

-> the

Changed in Line 17, 25 and 79

74 80 females **(4-6 years),

-> unnecessary repetition hinders clarity

individuals (4-6 years) of lidia cattle (184 males and 80 females),

74 belonging to 21 herds, selected according to criteria of genetic purity [6].

-> selection requires further explanation: does it refer to herds or to individuals? the "criteria of genetic purity" is not solved with a reference. This point is absolutely critical as the declared objective of the paper is the description of the breed, and it might be done on a selected sample!

We try to address your comment with this information in Lines 84 and 87 in combination with Reviewer 1 comments. As was stated for him the rest of the genetic lines described in some bibliographic reference for the breed are in extinction, without a sufficient number of adult animals, or are not currently kept in purity,  have been mixed with other lines. For this reason we have only sampled the “encastes” mentioned as representatives of the breed at present.

 belonging to 21 herds, selected as representatives herds in purity of the 15 genetic lines (called “encastes”) that are currently preserved of the breed: Miura, Pablo Romero, Veragua, Murube, Santa Coloma-Buendía, Santa Coloma -Graciliano, Gamero Cívico, Conde de la Corte, Atanasio-Lisardo, Domecq, Torrestrella, Núñez, Albaserrada, Vega-Villar and Navarra [6] were enrolled.

83 For the subsequent morphometric analysis photographs were **three-dimensionally converted

-> inappropriate or non-existant term

Changed in Line 117 and 118 to:

For the subsequent morphometric análisis, photographs were transformed into three dimensions files.

86 At the same time we have recorded

-> do they mean that these traits were recorded simultaneously?

Changed in Line 121:

In addition, complementary phaneroptic variables were recorded

88 In addition, the following **zoomometric indices

-> inappropriate or non-existant term

Changed to indexes in all the text.

103 presents the most variability is the length of the horn, especially in the female (C.V of 18.94%).

Changed, after English revision, in Lines 143 to 144 to:

The exterior length of the horn was less variable in males than females (p<0.05; Table 1).

109 Finally, Table 3 shows the percentage of males and females that present the categories of

-> are these significant differences? (this conclusion requires a two sample test's p-value for CV differences)

Animals were classified in each of the exposed categories. With the percentages by sex a contingency table was made and finally, in order to know if the distribution in males and females showed differences, we used the chi-square test (p<0,05), we hope that you consider it appropiate as well as the test you suggested. We have added all this in material and methods. 

110 morphological and phaneroptic variables studied.

-> unclear

Changed in Line 193 to 194 to:

Table 5 showed the different categories of morphological and phaneroptic variables studied depending on the gender.

112 In general we can consider the lidia breed as animals of mesolorian format within the bovine **specie.

-> This paragraph is very badly writen, plus I don't think this is the intended meaning for the word "specie".

Changed, after the English revision, the text is now in Lines 211 and 212

In general, the results obtained from body dimensions place the lidia breed as mid-sized compared across the bovine species [34]

Same for "dewlap", as its hipothesized relation with the dorsal-lumbar line requires further explanation or references.

Changed, after the English revision, the text is now in Lines 211 and 212

However, farmers selected males for variables of anterior third because the "badanudo" (developed dewlap) was a desired characteristic because contributes to increase the visual predominance of this third. For the current study, a developed dewlap was presented in 56.2% of males, whereas in the females, 48.1% of the cases had less developed dewlaps (Table 5).

132 female of Argentinean Creole cattle,

-> unjustified reference to a different breed. The context of breeds to compare belongs to the introduction.

This reference was suppresed.

134 The dorsal-lumbar line is observed tending more to saddling in the male

-> gramar

Changed in Line 235:.

The observed thoracolumbar line tended to be more saddled in the male

149 In the case of the head length (43.44 - 59.17 cm in the male),

-> is this the absolute range, or is it an inter-quantil? It must be stated here. An absolute range may be affected by outliers and should be avoided. The number of decimal digits is probably excesive for the precision of the set up.

This is the absolute range, maximun and minimun values of Table 1, outliers have been removed before performing statistical análisis and only one decimal is used now.

152 The average values obtained are lower than those described for the retinta breed [38], for the canarian

153 breed [39-40], for the avileña, limousin, saler and charolais breeds [41] and for the pallaresa breed

-> The authors don't provide a reason for comparing with these breeds.

-> Breeds in mixed cases, articles in mixed languages

There are autochthonous breed names that do not have an English translation (avileña, pallaresa)

We followed Reviewer 1 recomendation and mantain the comparation of Indexes with authoctonous breeds. The comparation with one or another breed is based on the information published in each paper, we dont have measurements of all the morphological parameters in all the breeds mentioned. Unfortunately there are not many published works on morphometry

187 brevile

-> unclear

Removed in the attempt to made discussion short and clear.

204 The anterior part predominates in the lidia breed [33], but there are genetic lines (encastes) that

The issue with "encastes" is critical and must be dealed with much earlier in this manuscript, and with greater detail.

In Material and Methods we have listed the different genetic lines (“encastes”) that we have sampled, therefore, here we can quote the genetic line in particular:

This text was removed in the attempt to made discussion short and clear.

215 idea of the sarcopoietic aptitude of each breed, considering the meatiness of a breed the more it

-> gramar

After English revision text in Lines 259 260 is:

provided an idea of the sarcopoietic aptitude of each breed, considering a better beef conformation for a breed when it exceeds the more from 50 points

222 borlon

-> language?

Removed in the attempt to made discussion short and clear.

229 breeds, as a result of the selection because the breeders have been selecting over the years bulls "short-legsl"

-> unclear

Changed, after the English revision, to Lines 266 and 267:

as a selection result because breeders have been selecting bulls with less height during the last century [47,48].

Height

238 for the bovine cattle of Uruguay

-> language

Changed, after the English revision, to Lines 257 and 258:

Similar values have been reported for Uruguay breeds

265 indexs

-> unclear

indexes

Reviewer 3 Report

The manuscript sent is very poor in all the aspects . The introduction lacks of references provided of the use of this methodology in other animal populations. I understand that this kind of methodology, if useful, would have to be widely used in wild populations. The lidia cattle population is extremely structures. Then, detailing the sampling of the herds is very important, as any study would be difficultly representative unless there are a lot ramdomly sampled herds, which is not probably the case (no information is provided in the paper).The recorded variables are not described. Instead, indexes coming from those are defined. The methodology is not described, just by very vaguely saying that one way analysis and matrix of correlations were computed. All the details are missing. From the results it is deduced that the only factor of variation included in the model is the sex. A single ttest would have been enough to do that. On the other hand, no other effect is included. I miss for example ages and herd effects to correct. More statistical analysis are missing, principal component analyses, for example. As the measures are taken roughly, a repeatability of them is needed. Results only concern basically means and comparisons form males to females, which is extremely poor for a scientist journal. Discussion only falls on differences between sexes almost trait by trait, but no implications are derived. The list of references very rarely are in papers published under peer review process and are normally written in Spanish. This gives an idea of the potential international interest of the research, or maybe of the poor search of papers carried out. My recommendation is clear to reject the paper.

Author Response

  1. The introduction lacks of references provided of the use of this methodology in other animal populations. I understand that this kind of methodology, if useful, would have to be widely used in wild populations.

From line 71 to 76 there is references to the photogrametry use in wild and domestic populations.

  1. The lidia cattle population is extremely structures. Then, detailing the sampling of the herds is very important, as any study would be difficultly representative unless there are a lot ramdomly sampled herds, which is not probably the case (no information is provided in the paper).

In Material and Methods Lines from 81 to 86 there is information about the different “encastes” or genetic lines enroled.

  1. The recorded variables are not described. Instead, indexes coming from those are defined.

The morphological variables are presented in Figure 2 and the References used are mentioned in Line 101. We decided this way to do it in an attept to no made the article excesive long, but definitions could be provided if Reviewer 3 still consider it necesary. Indexes are detailed from Lines 108 to 113 and also the references in Line 107.

  1. The methodology is not described, just by very vaguely saying that one way analysis and matrix of correlations were computed. All the details are missing.

New information is provided in Lines 114 to 121 .

  1. From the results it is deduced that the only factor of variation included in the model is the sex. A single ttest would have been enough to do that. On the other hand, no other effect is included.

We agree that T-test could be used for sex differences. Now new effect “encaste” is also considered in the principal component análisis Figure 3.

  1. I miss for example ages and herd effects to correct.

Animals were 4 to 6 years old, mature animals, and now “encaste” effect is include.

  1. More statistical analysis are missing, principal component analyses, for example.

Principal component análisis was done for males.

  1. As the measures are taken roughly, a repeatability of them is needed.

Each measure is made 5 times and the mean value of each one is used for the study. This is stated in Gaudioso et al. 2014 methodoloy. To repeat all this information could made the article excesively long but this information could be include in the pre

  1. Results only concern basically means and comparisons form males to females, which is extremely poor for a scientist journal.

Folllowing the other Reviewers recommendations some more Results are now provided.

  1. Discussion only falls on differences between sexes almost trait by trait, but no implications are derived.

Discussion is deeply changed folowing the recomendations of Reviewers.

  1. The list of references very rarely are in papers published under peer review process and are normally written in Spanish. This gives an idea of the potential international interest of the research, or maybe of the poor search of papers carried out.

We authors would love to provide more international published references, but this breed is difficult to study, both because of the inherent behavior of the animals, but also because accessing fighting farms is not as easy as other livestock farms.

Round 2

Reviewer 1 Report

RESULTS

Table 1. Eliminate significance of the coefficient of variation of the measure exterior length of the horn. Just mention that there is greater variability in females than in males.

The sentence “Lidia males were mesocephalic (50.6) while 154 females were slightly dolicocephalic (44.6, p<0.05; Table 2)”, is wrong. Cephalic index (CEI) refers to the harmony of the head, classifying the animals as brachycephalic (CEI > 100, prevalence of head width over head length), mesocephalic (CEI = 100, head length = head width), or dolichocephalic (CEI < 100, prevalence of head length over head width).

In the description in Table 2, only indicate which indices were different between males and females, and not mention what each one indicates, that is discussion.

Table 2. Write relative depth of thorax index and relative thickness of cannon bone index as previously agreed.

If an alpha of 0.05 was set to establish significant differences in the analysis of variance, this should not be said: “Proportionality index values were numerically less (p>0.10; Table 2) in males than females”, they are just statistically similar.

Table 3. The correlation located at the intersection of line 16 and column 15 is negative.

Table 4. The data is not consistent. The correlation located at the intersection of line 8 and column 1 has a coefficient equal to 0.18 and is considered statistically positive and significant, however there are multiple coefficients with a higher value and are not considered as such. The correlation located at the intersection of line 16 and column 15 is negative.

However, there is a greater inconsistency. The correlation analysis is very sensitive to the sample size, so when there is a larger sample size, statistical significance is found in coefficients of lower value. Therefore, it is not possible that in the case of males (n = 184) a coefficient of 0.19 is not significant and in the case of females (n = 80) a coefficient of 0.18 is significant. In female data, statistical significance should start in coefficients with a higher value.

Information must be reanalyzed to determine the number of significant positive correlations. It should also be indicated, between which body measurements the maximum correlations were observed, both in females and males.

Tables 3 and 4 can be unified as follows:

1

2

3

4

5

6

7

1

1.00

2

1.00

Data of

3

1.00

Females

4

1.00

5

Data of

1.00

6

males

1.00

7

1.00

In the description of Table 5 mention the differences between male and female of those measurements that were different according to the chi-square analysis.

The principal component analysis is incomplete. Two tables are missing, one with the eigenvalues and the explained variance percentage of each body measurement, and the other with the principal component factor scores coefficients. This is important because some principal components do not explain a significant proportion of variation and only those that account for approximately 80% of it are considered. It is also important to know that morphological measures are the cause of the variation within each of the components since some measures do not have any weight on it. Considering the above, figure 3 is not appropriate because it includes all body measurements and generally only a few are the causes of the total variation.

Looking at Figure 3, it is striking that the first two main components PC1 and PC2, explain only 58% of the total variance between the animals of the different “encastes”. If so, perhaps the data will have better quality for other types of analysis such as the canonical discriminant analysis. Box’s M, Kaiser-Meyer-Olkin or Bartlett's Sphericity tests is recommended to know what type of multivariate analysis the data fits. Likewise, Figure 3 shows a high morphological variability within some “encastes”, so perhaps the hierarchical cluster analysis would allow a better grouping of them.

DISCUSSION

Lines 305-315. The discussion is correct but it should consider all the measures that the principal component analysis considers relevant in order to achieve a better characterization of the similarities and differences of the different “encastes” studied.

It is recommended to carefully review the results of all the tables and especially to analyze the quality of the data to define which multivariate technique (main components, discriminant, canonical correlation) is the most suitable for data analysis. In addition, the full results of the multivariate analysis should be included and consider which type of technique allows a better graphic representation of the resulting groups (a score plot or a dendogram). Obviously the discussion and conclusions will have to be adjusted to the new presentation of results.

Reviewer 3 Report

The study aim is extremely poor as it is only measuring animals to describe a population from an anatomic point of view. The materials are completely not described and the methodology is also poorly described and it just refers to a very poor variance analysis with many factors missing as herd, "encaste", age, at least, but probably many others. The traits recorded are not defined.  The main conclusions of the paper are very poor and adds nothing important to the knowledge, since basically come to say that bullfighting cattle is smaller than the usual dairy and beef cattle breeds and that there is sexual dimorphism. A quick view to the references list, with testimonial JCR papers, and many of them in Spanish, inform about the narrow scope of the study. Again in my view the paper does not achieve the minimum standards to be published in a international scientific journal, and in my opion there is no room for improvement. Below there are some detailed comments that I expect to be useful for the authors.

Detailed comments

L84-86 How is the distribution of the number of records regarding "encastes"? Some results appearing later in the paper suggest that there is a very unbalance on them. Is this unbalance representative? Is it possible that some results later are conditioned for the low number of data of some "encastes"? How many herds are in each of the "encastes"?

L124-130 Authors insist in no describing the measures taken. So, how can a reader imagine how a cephalic index is for example computed? References of widths and lengths greatly affect the measures.

L132-133 I insist that one way analysis is a very poor analysis here. The only effect considered is the gender but what about the influence of age or herd (nested to "encaste") for example?

L135 Frequencies of what?

L136 What were the traits used in the principal component analysis? All those involved in computation of indexes would have to be studied together. Authors just say "principal component analysis without any specification. Rotations? What traits? What does it mean "encaste" in this context? Did the authors carry out the analysis within "encaste" or did they try to group individuals defining them according to grouping found in the analysis? I am not sure the authors really understand what they are doing.

L139-166 To my view, with the exception of the results of the analysis of variance, all this is part of the description of the materials and should be moved to the corresponding section.

L140-141 Authors discuss on the difference of the variability of traits between groups, but there is not a statistical procedure describe in the methodology to do that.

L148 Traits in table 1 have not been defined. I warned about this in my former revision, it is very important, and it has been ignored by the authors.

L149-150 No stars are in any trait in the m/f column? By the way SD is an acronym usually used for standard deviation.

L152 shows

L156 p>0.10? p values have to be defined in the design of an experiment, for example p<0.05, and then the effects are significant or not for the significance previously defined, (p<0.05). Many researchers change < to > when they do not find significance, bit this is an error. And greater error is to change the significance level to declare unsignificance.

L165 Does not exist sexual dimorphism in the indeces?

L189 This is the point in which the reader can see what traits have been analysed under chi-squared tests, when this would have to be stated in the methodology section. By the way, these traits were unknown for the reader to this point as they have not been defined in the materials section.

L197 the principal component analysis done is interesting but probably not the most interesting. The authors have recorded 20 traits, 6 indexes, and another around 10 discrete variables. The high genetic correlations between many of them suggest that some are probably redundant to describe the animals. The principal component analysis would have to add information about how many and what variables group together in each of the principal component factors, and mainly in the first one. The authors do not discuss on the high percentage of the variability gathered in the first component, reporting in addition that the second one only takes 9% of the total variance, suggesting that the whole variability is almost uniquely discriminated by 1 factor.

L212-213 I disagree. There are many species in the world non selected for anything with much higher sexual dimorphism.

L232-243 Disagree again. The selection objective in this breed has probably been behavior to fight. What the authors find in the morphology is in great extent a correlated genetic response.